# Lipidomic analysis of bile from patients with extrahepatic cholangiocarcinoma

**Seong Ji Choi[1], Hyo Jung Kim[1]\*, Sung Won Kwon[2]\*, Won Shik Kim[1], Jae Seon Kim[1]**

**1** Department of Internal Medicine, Korea University Guro Hospital, Korea University College of Medicine, Seoul, Republic of Korea, **2** College of Pharmacy, Seoul National University, Seoul, Republic of Korea

\* hjkimmd@korea.ac.kr (HJK); swkwon@snu.ac.kr (SWK)

## Abstract

### Objective

Cholangiocarcinoma and gallstones are significant gastrointestinal diseases with diverse etiologies and complex clinical manifestations. Understanding their underlying molecular mechanisms is crucial for advancing diagnosis and treatment strategies. In this study, we compared the lipidomic profiles of bile samples from patients with extrahepatic cholangiocarcinoma (eCCA) or choledocholithiasis with those of healthy controls to identify potential biomarkers and therapeutic targets.

### Methods

Bile samples were prospectively collected from 33 patients undergoing endoscopic retrograde cholangiopancreatography at the Korea University Guro Hospital, including 12 patients with eCCA, 15 with choledocholithiasis, and six controls. Lipidomic profiling was performed using ultra-high-performance liquid chromatography coupled with tandem mass spectrometry. Principal component analysis, ANOVA, and volcano plots were used to identify the differential lipidomic signatures across the groups.

### Results

A total of 230 lipid metabolites were identified, and significant differences were observed among the groups; each group had distinct lipidomic patterns. In both eCCA and choledocholithiasis, phosphatidylcholine contents were consistently more downregulated than in controls. However, diacylglycerol lipids were upregulated in eCCA while acylcarnitine lipids were upregulated in choledocholithiasis. Lysophosphatidylcholine levels were notably lower in patients with eCCA than in those with choledocholithiasis.

**Data availability statement:** All relevant data are within the paper and its Supporting information files.

**Funding:** The author(s) received no specific funding for this work.

**Competing interests:** The authors have declared that no competing interests exist.

## Conclusion

Our results suggested that specific lipidomic changes and their inter-relationships contribute to the pathophysiology of choledocholithiasis and eCCA. Longitudinal studies and functional assays can further validate the findings and translate them into clinical practice.

## Introduction

Cholangiocarcinoma (CCA) and gallstone disease present significant challenges in clinical practice owing to their diverse etiologies and complex pathophysiologies. CCA encompasses a group of heterogeneous malignancies originating from the bile duct epithelium, characterized by an aggressive clinical course and dismal prognosis [1]. It is mostly diagnosed at an advanced stage, severely limiting therapeutic options and significantly reducing patient survival rates.

Given the poor prognosis and limited treatment options associated with CCA, consistent research to improve the diagnostic and therapeutic approaches remains imperative. Key areas of focus include the bile-derived lipid profiles that could help identify potential biomarkers, enabling early diagnosis and intervention in CCA. Additionally, study of individual lipid variations can provide insights into the risk of gallstone formation and tumorigenesis in CCA, facilitating the development of person-alized treatments. Recently, lipidomics has emerged as an innovative and compre-hensive analytical approach that focuses on the characterization and quantification of lipids and their metabolic derivatives [2]. This approach can enhance diagnostic accuracy and guide targeted therapies by revealing the molecular pathways involved in CCA, eventually improving patient outcomes.

The current study aimed to investigate the lipidomic profiles of bile samples obtained from patients with extrahepatic cholangiocarcinoma (eCCA) using advanced mass spectrometry techniques. As bile duct obstruction can occur in both eCCA and choledocholithiasis, a comparative analysis of these diseases could help identify the distinct lipidomic signatures to differentiate one from the other. By comparing lipid compositions in bile samples from patients with eCCA and choledocholithiasis, we sought to uncover unique lipidomic patterns, providing valuable insights into the mechanisms underlying the disease and contributing to the advancement of future diagnostic and therapeutic strategies.

## Methods

### Study design and patient enrollment

Patients who underwent endoscopic retrograde cholangiopancreatography (ERCP) for eCCA or choledocholithiasis were prospectively enrolled between May 2022 and July 2023 at the Korea University Guro Hospital, Seoul, Republic of Korea. The control group included six patients who underwent ERCP with endoscopic nasobiliary drainage (ENBD) without biliary disease. The patients required bile drainage for bile duct injuries after hepatobiliary surgery. Those exhibiting symptoms of cholangitis,

presenting with turbid bile upon visual examination, or who were unable to maintain a normal diet were excluded from all groups.

Fresh bile (~ 5 mL) was aspirated 48–72 h after ENBD, once fever and abdominal pain had resolved, the patient had resumed oral intake, and the ENBD effluent was macroscopically clear and sludge-free, even though serum liver-function tests (LFTs) had not yet normalized. The collected bile samples were directly frozen at –80°C until further analysis.

The study protocol was approved by the Institutional Review Board of the Korea University Guro Hospital, Seoul, Republic of Korea (approval number: 2021GR0400). Written informed consent was obtained from all participants before enrollment.

## Lipidomic analysis

**Materials and equipment.** LC-MS-grade methanol, water (Merck, Darmstadt, Germany), methyl tert-butyl ether (MTBE), and toluene (99.9%, Sigma-Aldrich, St. Louis, MO, USA) were used for sample extraction. The internal standards included a spectrum of lipid molecules, such as lysophosphatidylcholines (LPC) 17:0, LPE 17:1, phosphatidylcholine (PC) 10:0, phosphatidylethanolamines (PE) 10:0, SM 17:0 (d18:1/17:0), sphingosine d17:1, diacylglycerol (DG) 20:1 (18:1/2:0), and MG 17:0 from Avanti polar lipids (Alabaster, AL, USA). The EquiSPLASH™ LIPIDOMIX™ Quantitative Mass Spec Internal Standard (Avanti Polar Lipids, Alabaster, AL, USA), containing 13 deuterated lipid internal standard, was also used for the standardization and quantification of lipid species in the samples.

For ultra-high-performance liquid chromatography (UHPLC) analyses, the solvent system comprised of LC-grade acetonitrile and 2-propanol (J. T. Baker, USA) and LC-MS-grade water (Merck, Darmstadt, Germany), with appropriate concentrations of formic acid and ammonium formate (LC-MS grade, Sigma-Aldrich, St. Louis, MO) serving as the salt components. The analytical equipment included a UHPLC (Agilent 1290 Infinity), time-of-flight mass spectrometer (Agilent Q-TOF 6530 MS), and a Q-Exactive Plus Hybrid Quadrupole-Orbitrap mass spectrometer (Thermo Fisher Scientific). The Waters ACQUITY CSH C18 (2.1 × 100 mm, 1.7 μm) column and Waters ACQUITY VanGuard CSH C18 1.7 μm pre-columns were used as LC separation columns.

**Lipid extraction and sample preparation.** Lipid extraction was conducted by dissolving the bile samples on ice at 0 °C, following the method of Matyash et al. (2008) [3]. Composition of the QC samples was determined by taking an equal amount of all samples in the experiment. The extraction process is described below.

For each 10 μL of bile sample, 300 μL of methanol at –20°C and 1000 μL of MTBE at –20°C were sequentially added and vortexed for 10 s. Then, the samples were extracted for 1 h at 4°C and 1500 rpm using a Thermo-shaker (ALL-SHENG). Next, 250 μL of water at 4°C was added, vortexed for 30 s, and centrifuged at 4 °C and 16000 RCF for 10 min using an Eppendorf 5415 C centrifuge. The 400-μL upper layer from the centrifuged and separated samples was transferred to an EP tube and dried using a Savant AES1010 SpeedVac (Thermo Fisher Scientific). Samples were reconstituted in 150 μL of MeOH:Toluene (9:1) solution prior to UHPLC coupled with tandem mass spectrometry (UHPLC-MS/MS) analysis.

**UHPLC-MS/MS analysis.** A 1-mg aliquot of each sample was analyzed using the Q-Exactive Plus Hybrid Quadrupole-Orbitrap mass spectrometer and the EquiSPLASH™ LIPIDOMIX® Quantitative Mass Spec Internal Standard kit, as described previously. The analysis utilized solvent A (water:ACN = 4:6, with 0.1% formic acid and 10 mM ammonium formate) and solvent B (IPA:ACN = 9:1, with 0.1% formic acid and 10 mM ammonium formate) as the mobile phases. The flow rate was maintained at 0.6 mL/min with the column temperature set at 65°C. The gradient program for solvent separation was set as follows: 15% B to 30% B from 0–2 min, 48% B from 2–2.5 min, 82% B from 2.5–11 min, 99% B from 11–11.5 min, held at 99% B for 12 min, then returned to 15% B at 12.1 min, and maintained there for 15 min, with an additional 4 min of post-run time. Data acquisition was performed in the data-dependent acquisition (DDA) full MS mode.

**Data analysis and quantification.** Raw data obtained from the analysis were processed using MS-DIAL (version 4.80) for peak selection, deconvolution, peak alignment, and compound identification. To validate the

results, manual peak identification and integration were performed using the LipidBlast library (version 68; https://fiehnlab.ucdavis.edu/projects/LipidBlast). Missing value imputation was performed using the K-Nearest Neighbor (KNN) algorithm via MetaboAnalyst 5.0, and data were filtered through interquartile range (IQR) filtering. Compounds with an RSD > 20% within the pooled QC samples were excluded from data interpretation. The concentration of each lipid species was determined relative to the peak intensity of an internal standard. The concentration values were log-transformed and subjected to Pareto scaling for visualization. Processed data were analyzed using multivariate statistical techniques, such as Principal Component Analysis (PCA), Analysis of Variance (ANOVA) tests, and hierarchical clustering heatmaps for statistical significance testing. GraphPad Prism (version 9.0) was used to graph the grouped data.

## Statistical analysis

For continuous variables, ANOVA was performed to evaluate the differences across the three groups. For categorical variables, a chi-square test was performed. The statistical analysis was conducted using IBM SPSS Statistics for Windows, Version 27.0 (IBM Corp., Armonk, NY, USA). Statistical significance was set at $p < 0.05$.

## Results

A total of 33 participants were enrolled in this study, including 12 patients with eCCA, 15 with choledocholithiasis, and six controls (Table 1). The mean age was significantly different across the groups ($p = 0.004$), and post-hoc analysis revealed that the control group was significantly younger. ALP and total bilirubin levels were significantly higher in the eCCA group than in the other two ($p < 0.001$ and $p = 0.001$, respectively). No statistically significant difference was observed in other parameters, including the proportion of male participants and AST, ALT, CRP, and CA 19−9 levels ($p > 0.05$). Detailed individual clinicopathological data of eCCA patients are provided in S1 Table. To enhance data clarity, box plots visualizing the distribution and variability of demographic and clinical parameters across the groups are presented in S1 Fig.

In the choledocholithiasis subgroup, we further reviewed stone characteristics based on endoscopic findings. Brown pigment stones were identified in 5 cases (33.3%), black pigment stones in 2 (13.3%), mixed brown–black stones in 3 (20.0%), and cholesterol stones in 2 (13.3%). Three cases (20.0%) exhibited indeterminate mixed pigment features. Because the number of patients in each subgroup was small and the composition was heterogeneous, no statistically meaningful comparison with bile lipid profiles could be performed.

**Table 1. Demographic and clinical characteristics of the study cohort.**

|  | Control (n = 6) | Choledocholithiasis (n = 15) | eCCA (n = 12) | p-value |
|---|---|---|---|---|
| **Age, yr** | 52.3 ± 13.3[a] (49.5) | 72.3 ± 14.2[b] (73.5) | 75.2 ± 11.5[b] (75.0) | 0.004 |
| **Male, n (%)** | 3 (50.0) | 8 (53.3) | 9 (75.0) | 0.437 |
| **AST** | 43.3 ± 38.4 (32.0) | 63.3 ± 40.6 (44.0) | 76.4 ± 63.0 (55.0) | 0.420 |
| **ALT** | 65.8 ± 73.7 (32.5) | 53.3 ± 21.3 (57.5) | 79.5 ± 61.3 (54.0) | 0.410 |
| **ALP** | 160.3 ± 164.6[a] (100.0) | 197.1 ± 91.1[a] (152.5) | 440.8 ± 207.0[b] (392.0) | < 0.001 |
| **Total bilirubin** | 2.7 ± 5.1[a] (0.6) | 2.0 ± 1.5[a] (1.3) | 10.3 ± 7.7[b] (7.4) | 0.001 |
| **CRP** | 68.1 ± 146.1 (8.5) | 68.4 ± 84.0 (40.3) | 40.1 ± 50.5 (28.7) | 0.680 |
| **CA 19−9** | 93.0 ± 181.93 (4.9) | 275.3 ± 397.3 (69.9) | 4340.5 ± 12050.0 (734.0) | 0.056 |

[a, b]Different superscripts indicate significant differences between groups ($p < 0.05$, Tukey's HSD post-hoc test).

Data are expressed as mean ± SD (median), except for male (n, %).

## Lipid metabolite detection

Using UHPLC-Q-Exactive Plus mass spectrometry, approximately 230 lipid metabolites were detected in bile samples. The identified lipid species included 21 triacylglycerides (TG), 20 DG, 82 PC, 17 LPCs, 14 PE, and 26 ceramides (Cer).

The PCA score plot (Fig 1) demonstrated distinct clustering among the eCCA, choledocholithiasis, and control groups. Notably, the eCCA samples clustered separately from the choledocholithiasis and control groups, highlighting their unique lipidomic profiles. Lipid concentrations in the bile from the eCCA group were generally lower than those in the other groups. The heatmap further supported these findings, demonstrating differential lipid expression in the eCCA group and suggesting a characteristic lipidomic signature. Additionally, the choledocholithiasis group displayed clustering of specific lipid subsets, reflecting distinct lipidomic patterns.

### Lipidomic differences between eCCA and control groups

Comparative lipidomic analysis using ANOVA revealed significant lipidomic differences between the eCCA and control groups. As shown in the volcano plot (Fig 2a), 13 PC-type lipids, including PC 37:5, PC 35:2, and PC 38:6, were significantly downregulated in the eCCA group. Conversely, two DG-type lipids, DG 38:9 and DG 32:1, were significantly upregulated in the eCCA group (fold change > 2, p < 0.01). The bar graph in Fig 2b illustrates the lipid alterations by comparing the relative abundances of significant lipid species between the eCCA and control groups. Most PC-type lipids were more

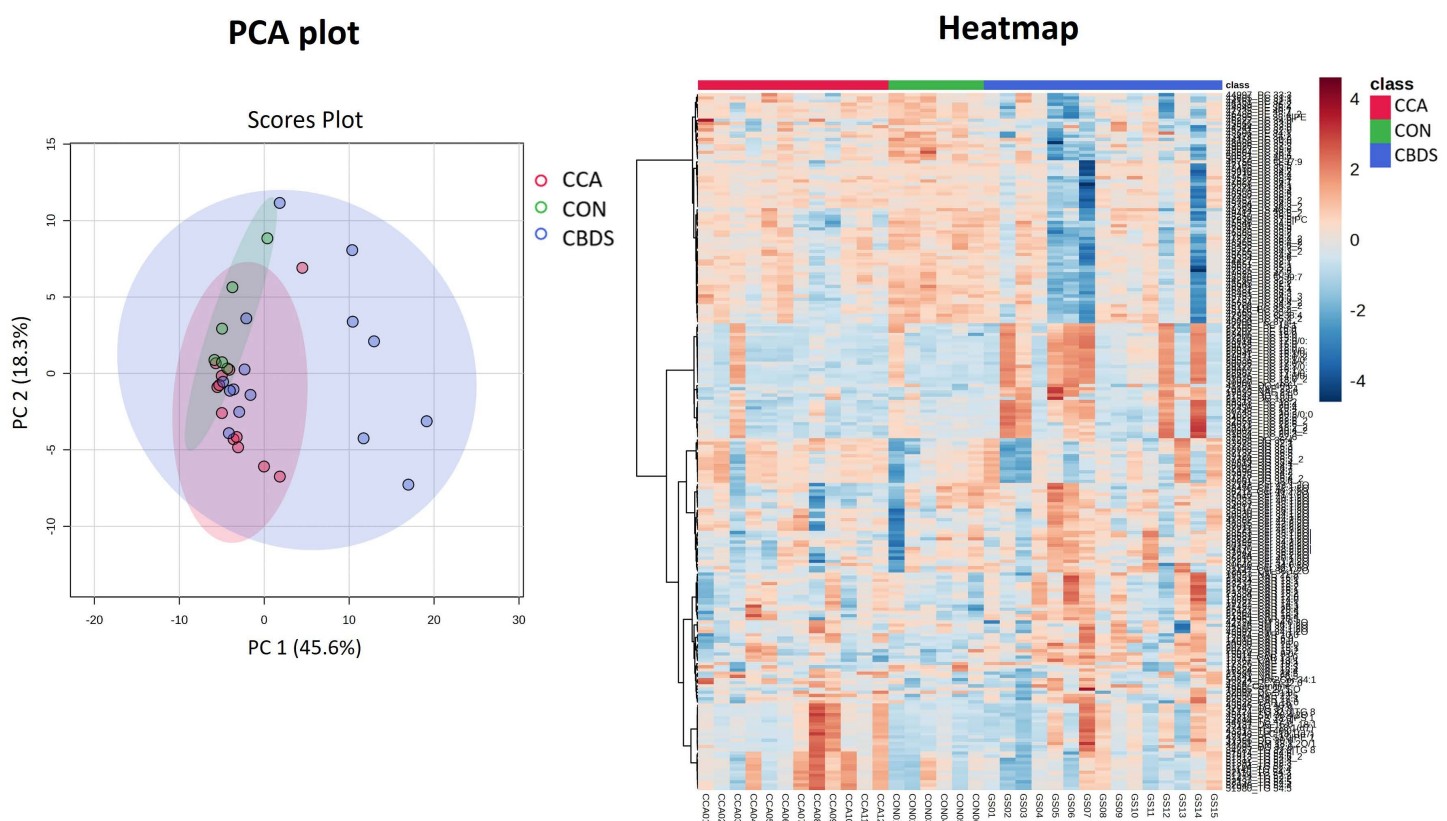

**Fig 1. PCA plot and heatmap revealing lipidomic signatures in extrahepatic cholangiocarcinoma (eCCA), choledocholithiasis (CBDS), and control (CON) groups.** These show the clustering of the experimental groups.

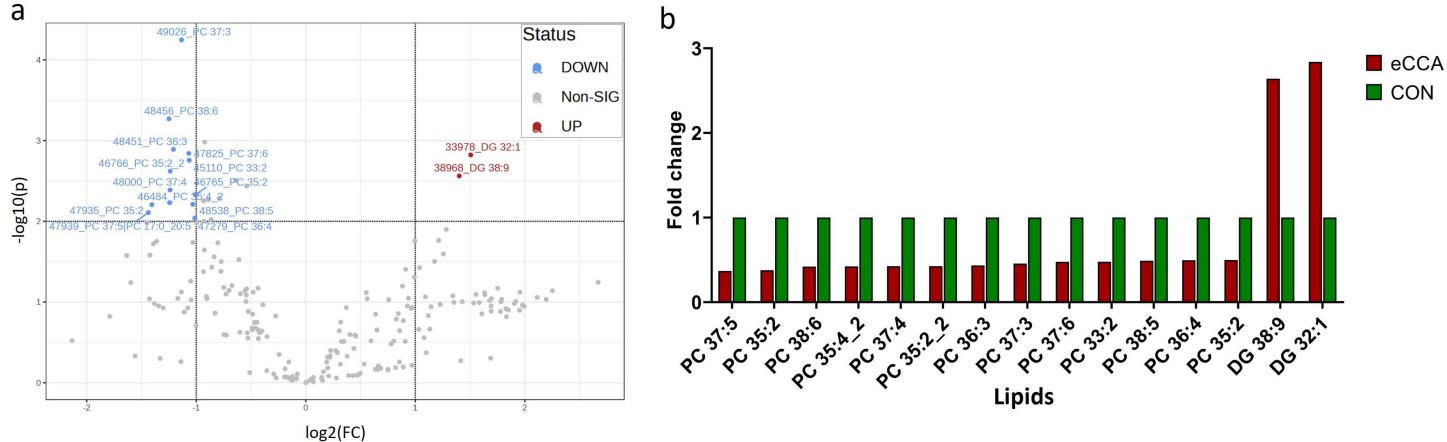

**Fig 2. Lipidomic analysis of bile samples comparing extrahepatic cholangiocarcinoma (eCCA) and control groups (CON) (fold change > 2; p-values<0.01). a.** Volcano plot and **b.** Bar graph. The eCCA group shows a reduction in 13 PC type lipids, and an increase in two DG type lipids.

abundant in the control group, whereas DG 38:9 and DG 32:1 were remarkably increased in the eCCA group. This pattern aligned well with the findings of the volcano plot, demonstrating the importance of these metabolites in differentiating the two groups.

## Lipidomic differences between choledocholithiasis and control groups

Lipidomic profiling revealed significant differences between the choledocholithiasis and control groups as demonstrated by ANOVA. According to the volcano plot (Fig 3a), PE 38:6 and 25 PC-type lipids, including PC 37:3, PC 38:6, and PC 36:5, were substantially downregulated in the choledocholithiasis group. Conversely, nine acylcarnitine (CAR)-type lipids, such as CAR 15:1, CAR 13:0, and CAR 8:0, were remarkably upregulated (fold change > 2, p < 0.01). Overall, the results revealed lipidomic disparities between the groups.

A bar graph (Fig 3b) presents the lipid changes, highlighting the pronounced elevation of CAR-type lipids in the choledocholithiasis group than in the control group. In contrast, the PC-type lipid levels were consistently higher in the control group. This lipidomic shift underscored the unique metabolic profile of the choledocholithiasis group and suggested that upregulated CAR-type lipids may serve as potential biomarkers for distinguishing patients with choledocholithiasis from healthy controls.

## Lipidomic differences between eCCA and choledocholithiasis groups

ANOVA revealed significant lipidomic differences between the eCCA and choledocholithiasis groups. The volcano plot in Fig 4a shows several lipids, notably LPC-type lipids, including LPC 16:0, LPC 17:0, and LPC 18:0, along with DG 40:6, to be significantly downregulated in the eCCA group (fold change < −2, p < 0.01). No significantly upregulated lipid was observed in the eCCA group compared to the choledocholithiasis group. The bar graph in Fig 4b illustrates the fold changes in these significant lipid species. The choledocholithiasis group consistently showed higher levels of several LPC-type lipids, whereas the eCCA group showed a notable decrease in the levels of these lipid species. This trend highlighted the reduced presence of LPC-type lipids in the eCCA group, consistent with the volcano plot findings. The concordance between the volcano plot and bar graph reinforced the potential role of downregulated LPC-type lipids as eCCA biomarkers.

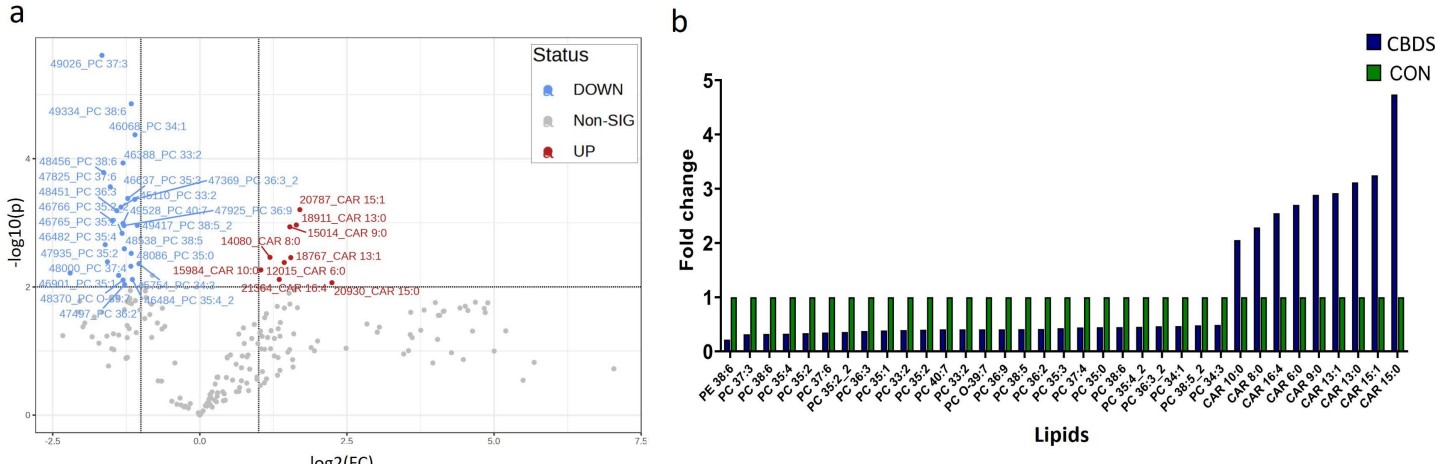

**Fig 3. Lipidomic analysis of bile samples comparing choledocholithiasis (CBDS) and control groups (CON) (fold change > 2; p-values<0.01). a.** Volcano plot and **b.** Bar graph. The CBDS group shows a significant decrease in 25 PC type lipids, but an elevation in 9 acylcarnitine type lipids.

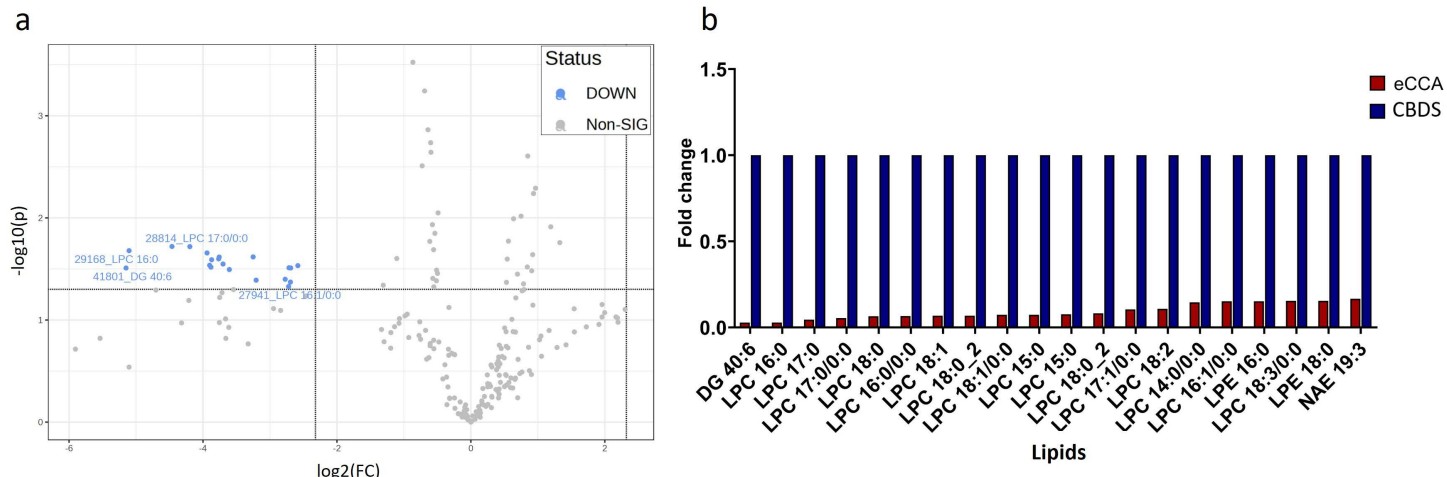

**Fig 4. Lipidomic analysis of bile samples comparing extrahepatic cholangiocarcinoma (eCCA) and choledocholithiasis (CBDS) groups (fold change > 5; p-values<0.05). a.** Volcano plot and **b.** Bar graph. There is the significant lipid difference in the eCCA group and CBDS group. CBDS group shows an increase in numerous LPC-type lipids.

## Discussion

The current lipidomic analysis described the lipidomic landscape distinguishing eCCA, choledocholithiasis, and control groups. The application of PCA, followed by detailed ANOVA tests, demonstrated significant alterations in bile lipid concentrations in patients with eCCA and choledocholithiasis compared with controls. The findings not only enhanced our understanding of the biochemical dynamics associated with eCCA and choledocholithiasis but also suggested new directions for identifying novel diagnostic biomarkers and therapeutic targets. The significant shifts in lipid profiles, graphically represented in volcano plots, offered a clear visualization of the lipidomic landscape associated with CCA, choledocholithiasis, and control conditions. In addition to changes in lipid profiles, their disequilibrium could contribute to gallstone formation and carcinogenesis.

Alterations in phosphatidylcholine (PC) metabolism emerged as a common feature of both disease groups. PC, also known as lecithin, constitutes as much as 40% of the organic material in bile and plays a crucial role in bile physiology. It supports the structural integrity and function of cell membranes, participates in signal transduction, and is essential for fat digestion and absorption [4–6]. Previous studies identified notable variations in the molecular composition of PC between the hepatic and biliary systems, pointing to the possibility of a specialized PC subset dedicated to bile secretion [7]. The biliary PC is physiologically significant in reducing the cytotoxic effects of bile salts through mixed micelle formation and protecting cholangiocytes from LPC-induced damage and carcinogenesis [5,8]. In the present study, a consistent down-regulation of multiple PC species—including PC 37:5, PC 35:2, and PC 38:6—was observed in both eCCA and choledo-cholithiasis compared to controls, suggesting that PC depletion may represent a shared pathophysiologic feature of biliary diseases. This reduction in PC levels may compromise bile micelle formation and cholangiocyte protection, potentially contributing to membrane instability, impaired bile secretion, and altered signal transduction.

Notably, both disease groups showed robust depletion of multiple PC species versus controls; however, the inter-disease difference did not reach the significance. In contrast, the eCCA group exhibited a unique enrichment in DG species, such as DG 38:9 and DG 32:1, whereas DG levels in choledocholithiasis remained comparable to controls. DG is an important membrane component and is known to act as a second messenger in various signal transduction pathways, particularly through the activation of protein kinase C (PKC) [9,10]. Given that PC and DG are metabolically intercon-vertible through phosphatidylcholine diacylglycerol choline phosphotransferase, the inverse relationship observed in PC depletion and DG elevation in eCCA may indicate a shift toward signaling-active lipid intermediates [11].

This metabolic "short-circuit" could favor pro-tumorigenic signaling in eCCA; however, this remains a hypothesis. Exper-imental loss of biliary PC secretion in *Abcb4(Mdr2)*-knock-out mice leads to ROS accumulation, DNA damage and chol-angiocarcinogenesis [12]. Also, multi-omics studies demonstrate that diacylglycerol-kinase-α (DGKA) is over-expressed in intrahepatic CCA and that pharmacologic DGKA blockade suppresses tumor growth [13,14]. Whether similar mechanisms operate in eCCA is currently unknown, and future validation experiments using human cholangiocyte models or animal systems targeting DGKA and related pathways are warranted.

Collectively, these findings suggest that early PC loss coupled with DG accumulation may act as a pathogenic driver rather than a mere consequence of eCCA development, although further mechanistic studies are needed. Interstingly, the findings align with studies from other cancers, including glioblastoma, where DG accumulation is associated with onco-genic signaling and cellular proliferation [15,16].

In contrast, choledocholithiasis displayed a distinct signature. While sharing the depletion of PC species, lipidomic profiles of patients with choledocholithiasis showed pronounced elevation of CAR-type lipids, including CAR 15:0 and CAR 15:1. CARs are esters of L-carnitine and fatty acids that play a crucial role in β-oxidation in mitochondria for energy metabolism, and their upregulation may reflect compensatory mechanism in response to bile acid imbalance or inflamma-tion [17,18]. Although the role of CAR in metabolic reprogramming related to stone formation remains unclear, the activa-tion of metabolic processes within cholangiocytes in response to the inflammatory reaction caused by bile duct stones and cholangitis could be the underlying cause. Experimental models assessing mitochondrial function and fatty acid oxidation in the setting of biliary inflammation may help clarify this mechanism.

Patients with choledocholithiasis also exhibit elevated levels of LPC, a proinflammatory lipid that causes cytotoxicity and induces oxidative DNA damage in cholangiocytes [19]. LPC, a major component of oxidatively damaged low-density lipoprotein (oxLDL), is generated through the hydrolysis of PC by phospholipase A2 (PLA2), often in response to biliary infection or pancreatic juice influx into the bile duct. By binding to G protein-coupled receptors and Toll-like receptors, LPC promotes the migration of lymphocytes and macrophages, increases the production of proinflammatory cytokines, such as interleukin-6 (IL-6) and tumor necrosis factor-alpha (TNF-α), induces oxidative stress, and triggers apoptosis [8,20,21]. Thus, the elevated LPC levels observed in choledocholithiasis may reflect an inflammatory cascade driven by PLA2 acti-vation in response to infection or cholestasis. The simultaneous rise of medium-chain CARs and saturated LPCs, together

with PC depletion, implies a self-reinforcing loop of mitochondrial β-oxidation overflow, membrane remodeling and disrupted bile homeostasis that precedes macroscopic stone formation.

The contrasting lipidomic signatures between patients with eCCA and choledocholithiasis highlight the distinct pathogenic mechanisms underlying the diseases despite some shared lipid alterations. For instance, while both conditions resulted in reduced PC levels, the differential upregulation of DG in the eCCA group and CAR/LPC in the choledocholithiasis group suggested unique metabolic adaptations specific to each disease state. The findings collectively emphasized the heterogeneity of biliary diseases and the need for tailored diagnostic and therapeutic approaches.

To our knowledge, very few studies on human fresh bile lipidome have been reported till date. Methodologically, the use of fresh bile samples in this study provided a more accurate reflection of the in-vivo bile lipidomic landscape than in previous studies that relied on concentrated bile from the gallbladder or bile duct [22]. By analyzing freshly secreted bile, we minimized artifacts introduced during bile concentration or storage, ensuring a reliable assessment of lipid composition in its natural state. The application of PCA and ANOVA further validated the robustness of our findings, demonstrating significant lipid alterations across the study groups. Visualization of these lipid shifts using volcano plots facilitated the identification of key lipid species that differentiated eCCA, choledocholithiasis, and control bile samples.

From a clinical perspective, these distinct lipidomic patterns offer potential for early diagnosis and therapeutic stratification. A combination of several different lipids, including PC, LPC, and CAR, may serve as an early indicator of cholangiocarcinoma while differentiating gallstone-related inflammation and metabolic dysfunction. Moreover, targeting lipid metabolic pathways, such as DG kinase alpha (DGKA), which regulates DG turnover, may represent a potential therapeutic strategy for eCCA. Given its role as a proliferation marker in intrahepatic CCA and its association with prognosis, DGKA may have similar implications in eCCA, warranting further investigation into its therapeutic and prognostic potential across CCA subtypes.

## Limitations

There are several limitations in this study. Due to the limited number of participants in the normal control group, the representativeness of each group should be taken into account. Significant differences in the baseline characteristics, including age and liver function tests, were observed between healthy controls and patients with eCCA and choledocholithiasis, which may have contributed to variability in bile lipid profiles. While statistical adjustment was limited by the small sample size, future studies with larger cohorts should incorporate covariate-adjusted analyses to address these potential confounders. Furthermore, as previous studies rarely specified the timing of ENBD bile collection, our sampling at 48–72 hours—after clinical resolution—was based on practical and physiological considerations. Due to ethical constraints, we used bile from patients with benign biliary injury as controls, rather than from truly healthy individuals. Although these controls had no malignancy or chronic biliary disease, baseline differences such as age and liver function may have introduced potential confounding effects.

## Conclusions

The lipidomic profiling study elucidated the metabolic disturbances associated with eCCA and choledocholithiasis, offering new insights into disease pathogenesis and potential avenues for early diagnosis. The observed elevation of DG species in eCCA possibly reflecting enhanced oncogenic signaling, warrants further experimental validation to assess causality. In choledocholithiasis, the concurrent increase in CAR-type lipids and LPC, along with a reduction in PC, reflects disrupted mitochondrial fatty acid oxidation and phospholipid remodeling, indicating a state of hepatobiliary metabolic stress. The distinct lipidomic signatures associated with CCA, compared to that in the choledocholithiasis and control groups, highlights the potential of lipids as biomarkers for disease diagnosis and progression monitoring. Future research should focus on mechanistic studies and longitudinal validation in larger cohorts to explore the diagnostic, prognostic, and therapeutic implications of bile lipidomics in biliary diseases.

## Supporting information

**S1 Table. Individual clinicopathological characteristics of patients with extrahepatic cholangiocarcinoma (n = 12).**
(DOCX)

**S1 Fig. Box plots of key clinical and laboratory variables across study groups.**
(TIF)

**S2 Fig. Raw data of demographic and clinical characteristics of the study cohort.**
(TIF)

**S3 Fig. Figure of significantly altered lipid species between extrahepatic cholangiocarcinoma (eCCA) and control (CON) groups.**
(TIF)

**S4 Fig. Figure of significantly altered lipid species between choledocholithiasis (CBDS) and control (CON) groups.**
(TIF)

**S5 Fig. Figure of significantly altered lipid species between extrahepatic cholangiocarcinoma (eCCA) and choledocholithiasis (CBDS) groups.**
(TIF)

## Acknowledgments

The authors declare that OpenAI's ChatGPT (version GPT-4.5, OpenAI, San Francisco, CA, USA) was used for language editing and improvement of readability. All scientific content, analysis, and interpretations are the sole responsibility of the authors.

## Author contributions

**Conceptualization:** Hyo Jung Kim, Sung Won Kwon.

**Data curation:** Won Shik Kim, Jae Seon Kim.

**Methodology:** Hyo Jung Kim, Sung Won Kwon.

**Writing – original draft:** Seong Ji Choi.

**Writing – review & editing:** Seong Ji Choi, Hyo Jung Kim, Sung Won Kwon, Won Shik Kim, Jae Seon Kim.

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
