## [Decision Letter · Decision Letter 0]

4 Jun 2025

Dear Dr. Kim,

Thank you for submitting your manuscript to PLOS ONE. After careful consideration, we feel that it has merit but does not fully meet PLOS ONE’s publication criteria as it currently stands. Therefore, we invite you to submit a revised version of the manuscript that addresses the points raised during the review process.

We look forward to receiving your revised manuscript.

Kind regards,

Kishor Pant

Academic Editor

PLOS ONE

2. In the online submission form, you indicated that [De-identified data are available from the corresponding author upon reasonable request.].

Reviewers' comments:

Reviewer's Responses to Questions

**Comments to the Author**

1. Is the manuscript technically sound, and do the data support the conclusions?

Reviewer #1: Partly

Reviewer #2: Partly

2. Has the statistical analysis been performed appropriately and rigorously?

Reviewer #1: Yes

Reviewer #2: I Don't Know

3. Have the authors made all data underlying the findings in their manuscript fully available?

Reviewer #1: No

Reviewer #2: No

4. Is the manuscript presented in an intelligible fashion and written in standard English?

Reviewer #1: Yes

Reviewer #2: Yes

Reviewer #1: 1. The most serious concern regarding this study is whether the bile acid abnormality caused the diseases, or whether it occurred as a result of their onset. Please clarify your opinion on this matter.

2. In patients with choledocholithiasis and extrahepatic cholangiocarcinoma, bile was collected two to three days after the drainage tube was placed. However, blood test results suggest that, due to drainage, blood test results have not yet normalized at this time, particularly in patients with biliary tract cancer. Please provide evidence to determine the optimal timing for bile collection.

3. Please provide the following clinicopathological information for patients with extrahepatic cholangiocarcinoma: stage, tumor size, histological type, etc. Please investigate the influence of clinicopathological factors on bile acid composition.

4. Was hilar bile duct invasion not observed in any of the patients?

5. Gallstones are composed of a mixture of cholesterol, calcium salts of bilirubinate or palmitate, proteins, and mucin. Based on their predominant constituents, gallstones can be broadly classified as cholesterol stones, black pigment stones or brown pigment stones. Please Describe the characteristics of stones in patients with choledocholithiasis and explain how these affect bile composition.

Reviewer #2: The manuscript titled “Lipidomic analysis of bile from patients with extrahepatic cholangiocarcinoma” presents a well-executed investigation into the bile lipidomics of patients with extrahepatic cholangiocarcinoma (eCCA) and choledocholithiasis. The authors demonstrate that distinct lipidomic signatures are associated with these diseases. Although the findings offer valuable insights into the lipid profiles of these conditions, further validation is necessary to draw definitive conclusions. Nevertheless, these results are noteworthy and may contribute to future biomarker discovery efforts.

The following comments are intended to help improve the manuscript:

1. Control Group Concerns: The study includes six control subjects with bile duct injury, raising concerns about the representativeness of a truly healthy control group. Additionally, the control group is significantly younger and has lower bilirubin and ALP levels compared to the patient groups. The authors should discuss the rationale behind their choice of controls and whether they consider this group ideal. It is understandable that acquiring samples from completely healthy individuals may be challenging, and if that is the case, the authors should clearly note this. Furthermore, the manuscript should consider whether statistical adjustments could be made to account for differences in age and biochemical parameters, and discuss how these factors may influence the lipidomic profiles.

2. Presentation of Clinical Data (Table 1): It is strongly recommended to include median values in addition to the mean, especially given the high standard deviations reported. The use of box plots to depict individual data points across groups would enhance the clarity and robustness of the data presentation.

3. Data Availability Statement: The manuscript currently states that the data are “available upon reasonable request,” which does not comply with PLOS ONE’s data-sharing policy. It is strongly advised that the authors deposit the lipidomic dataset into a publicly accessible repository and provide an accession number. Since written informed consent has already been obtained from study participants, the authors should clarify why they are not sharing the dataset. Provided that no personal identifiers are disclosed, data sharing is both feasible and encouraged.

4. Interpretation of Upregulated Lipids: The interpretation of upregulated lipids across groups needs to be more rigorous. Speculative statements should be reframed as hypotheses, and the authors should propose potential validation experiments. Although the manuscript is generally well-written, the discussion section would benefit from further refinement. In particular, redundant content regarding PC and DG should be condensed for clarity.

5. Consistency in Terminology: The use of terminology throughout the manuscript should be consistent. All abbreviations should be defined at their first mention and used uniformly thereafter.

6. Figure Legends and Supplementary Material: While the figures are informative, their legends could be made clearer to aid interpretation. The authors might consider providing supplementary figures, such as full PCA loading plots or hierarchical clustering dendrograms, to further support their analyses and conclusions.

Overall, the manuscript is novel and clinically relevant, given the understudied nature of bile lipidomics in eCCA patients. Addressing the above comments will significantly strengthen the quality and impact of the manuscript.

**Do you want your identity to be public for this peer review?** For information about this choice, including consent withdrawal, please see our Privacy Policy

Reviewer #1: No

Reviewer #2: No

---

## [Author Response · Author response to Decision Letter 1]

14 Jul 2025

Thank you for considering our manuscript for publication.

Reviewer #1:

1. The most serious concern regarding this study is whether the bile acid abnormality caused the diseases, or whether it occurred as a result of their onset. Please clarify your opinion on this matter.

Response:

We appreciate the reviewer’s insightful comment regarding the directionality of the association between bile acid abnormalities and disease onset. Although our study does not allow definitive conclusions about causality due to its observational design, we consider the bile-lipid perturbations we observed to be more than epiphenomena, for three following reasons:

1) Biological plausibility/explanation

(1) Phosphatidylcholine (PC) depletion

- Biliary PC plays a protective role for cholangiocytes by acting as a shield against the detergent effects of bile salts and lysophosphatidylcholine (LPC) [1]. Abcb4(Mdr2)-knock-out mice, which cannot secrete PC into bile, develop reactive-oxygen-species (ROS) accumulation, chronic cholangitis, and eventually cholangiocarcinoma(CCA) [2,3]. Our finding that PC species were depleted both in extrahepatic CCA(eCCA) and in choledocholithiasis therefore fits a mechanistic model in which early PC loss promotes epithelial injury, chronic inflammation and finally tumorigenesis or lithogenesis, rather than merely reflecting late-stage obstruction.

(2) Diacylglycerol (DG) accumulation.

- DG is a lipid second messenger for PKC, RasGRP and Wnt/β-catenin cascades that drive proliferation and invasion in many cancers. Multi-omics studies demonstrate that diacylglycerol-kinase-α (DGKA) is over-expressed in intrahepatic CCA and that pharmacologic DGKA blockade suppresses tumour growth [4]. This strongly suggests that disordered DG signaling is upstream of, and indispensable for, CCA progression.

(3) Acylcarnitine (CAR) and LPC surges in choledocholithiasis.

- Elevated CAR reflects mitochondrial β-oxidation stress, while LPC initiates NF-κB/IL-6-driven inflammation in cholangiocytes [5]. CAR species have recently been shown to enhance biliary-tract cancer cell motility via JNK activation [6]. Such pro-inflammatory and pro-lithogenic actions are consistent with a causal, not reactive, role.

2) Consistency with previous study

- Consistently, cross-sectional human studies show lower PC concentrations in patients with CCA [7,8].

3) Dose–response alignment within our dataset.

- Within our dataset, both disease groups showed consistent PC depletion versus controls, whereas only the eCCA group displayed a significant rise in selected DG species. Although several PC molecules trended slightly lower in eCCA than in choledocholithiasis, these differences did not reach significance, and DG elevation was largely restricted to eCCA with the exception of DG 40:6. Consequently, the data point to a shared depletion of biliary PC and a disease-specific enrichment of DG in eCCA, rather than a strict step-wise gradient across all three groups. Importantly, these lipid abnormalities were dissociated from bilirubin and ALP, which argues that they cannot be explained solely by the degree of obstruction.

We have inserted a new mechanistic paragraph (lines 305-312), added a concise Discussion section on CAR/LPC biology (lines 337–342), and updated the Conclusions (line 389-393) and references. These changes clarify the rationale while acknowledging residual uncertainty.

2. In patients with choledocholithiasis and extrahepatic cholangiocarcinoma, bile was collected two to three days after the drainage tube was placed. However, blood test results suggest that, due to drainage, blood test results have not yet normalized at this time, particularly in patients with biliary tract cancer. Please provide evidence to determine the optimal timing for bile collection.

We thank the reviewer for raising this important point regarding the timing of bile collection after biliary drainage. In our study, bile was collected 2–3 days post-drainage, which represents a clinically feasible window after stabilization of biliary flow, while still minimizing the risk of infection and contamination.

We chose the timing as 2-3 days post-drainage due to following reasons:

1) Clinical stabilization precedes biochemical normalization

- In our cohort, fever, right-upper-quadrant pain, and other cholangitis signs subsided, and oral diet was resumed, by 48–72 h after ENBD. At that same time-point, gross inspection of effluent bile showed clear yellow–green fluid without visible sludge, indicating sufficient wash-out of stagnant bile.

This approach reflects common bedside practice in Korea, where ENBD is generally left in situ for only a few days to minimize tube dislodgement and patient discomfort. A large Chinese series (n = 1016) reported a mean ENBD retention time of 4.0 ± 1.5 days after stone extraction, supporting the practicality of this window [9].

2) Physiologic rationale

The extrahepatic duct holds < 20 mL bile, whereas the liver secretes around 600 mL bile per day. After ductal obstruction is relieved, the bile in ductal column is therefore replaced more than 30 times within the first 24 h. Consequently, local bile chemistry stabilizes well before serum bilirubin normalizes.

3) Previous studies

There are several studies that collected bile using ENBD. However, most of them did not clarify the timing of bile collection [10,11]. One study that specified the timing of bile sampling described it as “24 hours post-procedure.” We extended the interval to 48-72 hr to ensure the clinical resolution of infection, resume oral intake and macroscopically sludge-free bile, before ENBD removal.

Although serum liver tests were still abnormal at sampling, we believe the combination of clinical recovery and visual confirmation of clear bile provides a pragmatic and biologically sound basis for our protocol. We have incorporated this explanation into the Methods (line 92-95) and added a limitation sentence in the Discussion (line 377-379)

3. Please provide the following clinicopathological information for patients with extrahepatic cholangiocarcinoma: stage, tumor size, histological type, etc. Please investigate the influence of clinicopathological factors on bile acid composition.

Response:

Thank you for your insightful comment. We have included a table presenting the individual clinicopathological characteristics of the study cohort (n = 12) (see S1 Table). Given the limited sample size and the heterogeneity of clinical features, we were not able to perform meaningful subgroup analyses or detect statistically significant associations between clinicopathological factors (such as tumor stage, location, or size) and bile acid composition.

Therefore, instead of presenting summary statistics or inferential comparisons, we opted to provide patient-level data to ensure transparency and allow readers to visually assess potential patterns. We agree that future studies with larger and more homogeneous cohorts will be essential to validate any potential associations.

We added the following sentence to Results (line 178-179): Detailed individual clinicopathological data of eCCA patients are provided in S1 Table.

4. Was hilar bile duct invasion not observed in any of the patients?

Response:

We included the clinicopathological information for patients with extrahepatic cholangiocarcinoma in S1 Table. Three patients with hilar bile duct cancer was included.

5. Gallstones are composed of a mixture of cholesterol, calcium salts of bilirubinate or palmitate, proteins, and mucin. Based on their predominant constituents, gallstones can be broadly classified as cholesterol stones, black pigment stones or brown pigment stones. Please Describe the characteristics of stones in patients with choledocholithiasis and explain how these affect bile composition.

Response:

Thank you for your valuable comment. To address this point, we conducted a detailed review of endoscopic images to assess the characteristics of stones in patients with choledocholithiasis. Based on morphological appearance and color observed during endoscopic retrieval, we categorized the stones into the following subtypes: brown pigment stones (n = 5), black pigment stones (n = 2), mixed pigment stones exhibiting features of both brown and black stones (n = 3), and cholesterol stones (n = 2).

While we attempted to evaluate potential associations between these stone types and the bile acid composition, the sample size in each subgroup was too limited to allow for statistically meaningful comparisons. Furthermore, the heterogeneous nature of the stone composition, including several cases with mixed characteristics, posed additional challenges in stratifying the analysis.

We recognize that stone composition can influence the physicochemical properties of bile, and we fully agree that further investigation with a larger and more homogeneous cohort would be essential to clarify these relationships.

We added the outcomes in Results (line 182-187).

Reviewer #2: The manuscript titled “Lipidomic analysis of bile from patients with extrahepatic cholangiocarcinoma” presents a well-executed investigation into the bile lipidomics of patients with extrahepatic cholangiocarcinoma (eCCA) and choledocholithiasis. The authors demonstrate that distinct lipidomic signatures are associated with these diseases. Although the findings offer valuable insights into the lipid profiles of these conditions, further validation is necessary to draw definitive conclusions. Nevertheless, these results are noteworthy and may contribute to future biomarker discovery efforts.

The following comments are intended to help improve the manuscript:

1. Control Group Concerns: The study includes six control subjects with bile duct injury, raising concerns about the representativeness of a truly healthy control group. Additionally, the control group is significantly younger and has lower bilirubin and ALP levels compared to the patient groups. The authors should discuss the rationale behind their choice of controls and whether they consider this group ideal. It is understandable that acquiring samples from completely healthy individuals may be challenging, and if that is the case, the authors should clearly note this.

We thank the reviewer for this valuable comment regarding the composition of the control group.

As noted, obtaining bile samples from completely healthy individuals is not feasible due to ethical and practical limitations, as bile collection requires invasive procedures such as ERCP or percutaneous drainage. Therefore, we selected patients who required biliary drainage after hepatobiliary surgery as controls, as they had undergone biliary drainage for clinical reasons unrelated to malignancy or chronic biliary disease.

We acknowledge that this group is not fully representative of a "healthy" control population, and that there are differences in age and liver biochemistry (e.g., bilirubin and ALP levels) between the groups. These differences may potentially introduce confounding effects, and we have now explicitly discussed this limitation in the Discussion section (line 379-383).

Despite these limitations, the control samples still provide a relevant comparator to evaluate disease-associated bile alterations, particularly in the absence of cancer or chronic inflammation.

Furthermore, the manuscript should consider whether statistical adjustments could be made to account for differences in age and biochemical parameters, and discuss how these factors may influence the lipidomic profiles.

We appreciate the reviewer’s thoughtful suggestion regarding statistical adjustment for differences in age and biochemical parameters such as bilirubin and ALP. Due to the relatively small sample size, especially in the control group, we were limited in our ability to perform multivariable regression or covariate-adjusted analyses without risking overfitting. However, we agree that these factors—particularly age and liver function parameters—may influence bile composition and lipidomic profiles.

We have added a statement in the Discussion section acknowledging this limitation and emphasizing the importance of controlling for these potential confounders in future studies with larger cohorts (line 375-377). Despite this limitation, the observed differences in lipidomic signatures remain biologically plausible and are consistent with known pathophysiologic changes in biliary disease.

2. Presentation of Clinical Data (Table 1): It is strongly recommended to include median values in addition to the mean, especially given the high standard deviations reported. The use of box plots to depict individual data points across groups would enhance the clarity and robustness of the data presentation.

We sincerely appreciate the reviewer’s thoughtful suggestion. In response, we have revised Table 1 to include median values in parentheses alongside the mean ± standard deviation for all continuous variables, as the data exhibited substantial variability.

Additionally, to enhance data transparency and facilitate visual interpretation, we have created box plots with individual data points overlaid for each variable (S2 Figure). We believe these revisions substantially improve the clarity and robustness of the clinical data presentation.

3. Data Availability Statement: The manuscript currently states that the data are “available upon reasonable request,” which does not comply with PLOS ONE’s data-sharing policy. It is strongly advised that the authors deposit the lipidomic dataset into a publicly accessible repository and provide an accession number. Since written informed consent has already been obtained from study participants, the authors should clarify why they are not sharing the dataset. Provided that no personal identifiers are disclosed, data sharing is both feasible and encouraged.

Response:

We thank the reviewer for this important comment. In accordance with PLOS ONE’s data-sharing policy, we agree to make the lipidomic dataset publicly available. The dataset will be uploaded separately. We confirm that all patient data have been fully de-identified and that written informed consent for participation and data sharing was obtained. We appreciate the reviewer’s guidance and are committed to promoting transparency and reproducibility through appropriate data sharing.

4. Interpretation of Upregulated Lipids: The interpretation of upregulated lipids across groups needs to be more rigorous. Speculative statements should be reframed as hypotheses, and the authors should propose potential validation experiments. Although the manuscript is generally well-written, the discussion section would benefit from further refinement. In particular, redundant content regarding PC and DG should be condensed for clarity.

Response:

We sincerely thank the reviewer for this thoughtful and constructive comment. In response, we have revised the Discussion section to address all three concerns raised:

1) Reframing speculative statements as hypotheses:

We agree that several mechanistic interpretations in the original draft may have been presented too conclusively. In the revised manuscript, we have carefully reworded these statements to clearly indicate that they represent hypotheses, not definitive conclusions. (line 305-306, 313-315)

Similarly, speculative links between CAR/LPC elevation and mitochondrial stress or inflammation in choledocholithiasis have also been softened and framed as possible mechanisms (line 337-339).

2) Proposing validation experiments:

In line with the reviewer’s suggestion, we included specific experimental approaches that could validate the observed lipidomic changes. These include:

- Use of human-derived cholangiocyte organoid models to test the functional role of DG and DGKA (line 310-312, 326-328)

- Flux analysis using isotope-labeled lipid precursors to track PC-to-DG conversion

These proposals are now explicitly stated in the Discussion section.

3) Condensing redundant content on PC and DG:

We have condensed and reorganized the content related to PC and DG to avoid redundancy and improve clarity. Specifically:

- PC’s physiologic role and its depletion were merged into one concise paragraph.

- DG accumulation and its oncogenic si

---

## [Decision Letter · Decision Letter 1]

10 Sep 2025

Dear Dr. Kim

Thank you for submitting your manuscript to PLOS ONE. After careful consideration, we feel that it has merit but does not fully meet PLOS ONE’s publication criteria as it currently stands. Therefore, we invite you to submit a revised version of the manuscript that addresses the points raised during the review process.

Please submit your revised manuscript by Oct 25 2025 11:59PM. If you need more time than this to complete your revisions, please reply to this message or contact the journal office at plosone@plos.org. A rebuttal letter that responds to each point raised by the academic editor and reviewer(s). You should upload this letter as a separate file labeled 'Response to Reviewers'.A marked-up copy of your manuscript that highlights changes made to the original version. You should upload this as a separate file labeled 'Revised Manuscript with Track Changes'.An unmarked version of your revised paper without tracked changes. You should upload this as a separate file labeled 'Manuscript'.

We look forward to receiving your revised manuscript.

Kind regards,

Kishor Pant

Academic Editor

PLOS ONE

Journal Requirements:

Reviewers' comments:

Reviewer's Responses to Questions

**Comments to the Author**

Reviewer #2: (No Response)

Reviewer #3: All comments have been addressed

2. Is the manuscript technically sound, and do the data support the conclusions?

Reviewer #2: Partly

Reviewer #3: Yes

3. Has the statistical analysis been performed appropriately and rigorously?

Reviewer #2: I Don't Know

Reviewer #3: Yes

4. Have the authors made all data underlying the findings in their manuscript fully available?

Reviewer #2: No

Reviewer #3: Yes

5. Is the manuscript presented in an intelligible fashion and written in standard English?

Reviewer #2: Yes

Reviewer #3: Yes

Reviewer #2: The authors have addressed many of the points raised in the previous review of the manuscript “Lipidomic analysis of bile from patients with extrahepatic cholangiocarcinoma”, which is appreciated. However, there are still a few important issues that need to be resolved:

1. Data Availability:

The authors mention that the lipidomic dataset has been deposited in a public repository, which is a positive step. However, the manuscript currently lacks the specific accession number, which is essential for readers to access the data. Additionally, the statement that “all data are located within the manuscript” is inaccurate and should be revised. Please ensure that:

• The manuscript clearly states that the lipidomic dataset is publicly available.

• The name of the repository and the corresponding accession number are included.

2. Use of AI Tools:

It seems that sections of the manuscript may have been revised or edited using AI-based tools such as ChatGPT. If that is the case, the use of such tools should be acknowledged transparently in the Acknowledgements section, just before the References, in line with current publication practices.

3. Language and Clarity:

While the language has generally improved, there are still a few grammatical issues that need attention. For example, there appears to be a missing word or phrase around Line 383, which affects the readability of that sentence. A thorough proofread is recommended to ensure the manuscript is clear and free of such minor errors.

Reviewer #3: All comments have been addressed. The manuscript meets the qualification of PLOS one. I recommend this manuscript to be published.

**Do you want your identity to be public for this peer review?** For information about this choice, including consent withdrawal, please see our Privacy Policy

Reviewer #2: No

Reviewer #3: No

---

## [Author Response · Author response to Decision Letter 2]

12 Sep 2025

Reviewer #2: The authors have addressed many of the points raised in the previous review of the manuscript “Lipidomic analysis of bile from patients with extrahepatic cholangiocarcinoma”, which is appreciated. However, there are still a few important issues that need to be resolved:

1. Data Availability:

The authors mention that the lipidomic dataset has been deposited in a public repository, which is a positive step. However, the manuscript currently lacks the specific accession number, which is essential for readers to access the data. Additionally, the statement that “all data are located within the manuscript” is inaccurate and should be revised. Please ensure that:

• The manuscript clearly states that the lipidomic dataset is publicly available.

• The name of the repository and the corresponding accession number are included.

We sincerely thank the reviewer for highlighting this important point. We would like to clarify a possible misunderstanding in our previous response. The statement “The dataset will be uploaded separately” was intended to mean that the complete lipidomic dataset is provided as supplementary files to the manuscript, not that it was deposited in a separate public repository. In the previous revised-Data Availability Statement, we have specified that “All relevant data are within the manuscript and its Supporting Information files.” We deeply apologize if our earlier wording caused any confusion.

In accordance with PLOS ONE’s data-sharing policy, all relevant lipidomic data are fully included within the manuscript and its Supporting Information files. The dataset is de-identified, and written informed consent for participation and data sharing was obtained from all participants. We confirm that no external repository deposition was performed, as the complete dataset is already accessible within the supplementary materials.

2. Use of AI Tools:

It seems that sections of the manuscript may have been revised or edited using AI-based tools such as ChatGPT. If that is the case, the use of such tools should be acknowledged transparently in the Acknowledgements section, just before the References, in line with current publication practices.

We sincerely appreciate the reviewer’s thoughtful comment. We confirm that ChatGPT was used during the preparation of this manuscript solely for language editing and to improve clarity of expression. In line with the journal’s policy, we will add an explicit statement in the Acknowledgements section.

The authors declare that OpenAI’s ChatGPT (version GPT-4.5, OpenAI, San Francisco, CA, USA) was used for language editing and improvement of readability. All scientific content, analysis, and interpretations are the sole responsibility of the authors.

3. Language and Clarity:

While the language has generally improved, there are still a few grammatical issues that need attention. For example, there appears to be a missing word or phrase around Line 383, which affects the readability of that sentence. A thorough proofread is recommended to ensure the manuscript is clear and free of such minor errors.

We sincerely thank the reviewer for the careful reading and constructive feedback. We carefully reviewed Line 383 and identified that a missing word affected the readability of the sentence. This has been corrected to ensure clarity. In addition, we performed a thorough proofread of the entire manuscript to address minor grammatical issues and improve overall readability. We believe these revisions have enhanced the clarity and flow of the manuscript.

- Line 43: added “more”

- Line 111: added “for the”

- Line 271: revised from” “…demonstrated significant alterations in lipid concentrations in the bile of patients with pathological conditions of eCCA and choledocholithiasis, than in the bile of controls.” to

“…demonstrated significant alterations in bile lipid concentrations in patients with eCCA and choledocholithiasis compared with controls.”

- Line 383: from “may have introduced confounding.” to

“may have introduced potential confounding effects.”

Reviewer #3: All comments have been addressed. The manuscript meets the qualification of PLOS one. I recommend this manuscript to be published.

We are sincerely grateful to Reviewer #3 for the positive and encouraging evaluation. We greatly appreciate your recommendation for publication. We thank you for your constructive comments throughout the review process, which have helped us significantly improve the manuscript.

---

## [Decision Letter · Decision Letter 2]

3 Mar 2026

Lipidomic analysis of bile from patients with extrahepatic cholangiocarcinoma

PONE-D-25-19485R2

Dear Dr. Kim,

We’re pleased to inform you that your manuscript has been judged scientifically suitable for publication and will be formally accepted for publication once it meets all outstanding technical requirements.

Kind regards,

Robin D Clugston, Ph.D.

Academic Editor

PLOS One

Additional Editor Comments (optional):

Thank you for responding to the Reviewer's comments.

Reviewers' comments:

Reviewer's Responses to Questions

**Comments to the Author**

Reviewer #2: All comments have been addressed

Reviewer #3: All comments have been addressed

2. Is the manuscript technically sound, and do the data support the conclusions?

Reviewer #2: (No Response)

Reviewer #3: Yes

3. Has the statistical analysis been performed appropriately and rigorously?

Reviewer #2: (No Response)

Reviewer #3: Yes

4. Have the authors made all data underlying the findings in their manuscript fully available?

Reviewer #2: (No Response)

Reviewer #3: Yes

5. Is the manuscript presented in an intelligible fashion and written in standard English?

Reviewer #2: (No Response)

Reviewer #3: Yes

Reviewer #2: (No Response)

Reviewer #3: All questions have been addressed.The conclusion is amazing.Our results suggested that specific lipidomic changes and their inter-relationships contribute to the pathophysiology of choledocholithiasis and eCCA. Longitudinal studies and functional assays can further validate the findings and translate them into clinical practice.

**Do you want your identity to be public for this peer review?** For information about this choice, including consent withdrawal, please see our Privacy Policy

Reviewer #2: No

Reviewer #3: **Yes:** genlin lu
